# The Physiotherapy Process of a Plegic Patient Who Communicates with Foot Movement—A Case Report

**DOI:** 10.3390/brainsci12060688

**Published:** 2022-05-25

**Authors:** Krzysztof Głowacki, Daniel Malczewski, Karolina Krzysztoń, Aniela Jasińska, Izabela Domitrz

**Affiliations:** 1Department of Neurology, Bielański Hospital, 01-809 Warsaw, Poland; 2Department of Neurology, Faculty of Medical Sciences, Medical University of Warsaw, 01-809 Warsaw, Poland; dmalczewski@wum.edu.pl (D.M.); izabela.domitrz@wum.edu.pl (I.D.); 3“Konstancja” Respiratory Ventilation Clinic, 05-520 Konstancin-Jeziorna, Poland; jasinskaaniela@gmail.com

**Keywords:** acute encephalitis, comprehensive rehabilitation, triplegia, case report

## Abstract

There are no official recommendations regarding physiotherapy for encephalitis patients. However, such patients, depending on their condition, have to undergo rehabilitation preceded by a detailed functional examination. The paper describes the physiotherapy treatment of a 28-year-old female after acute encephalitis. She suffered three-limb palsy with preserved movement in the right ankle joint. The patient was admitted to a clinic that offers respiratory therapy, where she underwent a comprehensive rehabilitation process. The initial and final functional assessment was conducted based on the International Classification of Functioning, Disability and Health. The therapy aimed to enable social contacts by learning to communicate with the environment, taking advantage of learned motor skills and adapting the body to maintain a sitting position. The goals were implemented with gradual upright standing, electrostimulation of paralyzed muscles, orofacial therapy, methods of respiratory acceleration and the use of communication technologies. As a result of the physiotherapy, the patient can communicate with the environment more efficiently and showed a more assertive attitude towards the disease and greater motivation to exercise and socialize. This paper supports the importance of a rehabilitation program adapted to the needs of a patient with severe disabilities and encourages more studies in this area.

## 1. Introduction

Acute encephalitis is a sudden-onset disease that affects about 7 out of 100,000 people in a population. The disease is characterized by behavioral disturbances, fever, seizures, and neurological deficits [1]. The most common causes of acute encephalitis include infections and autoimmune infections. However, a large proportion (37–62%) of cases remain classified as ones of unknown etiology. The common late diagnosis of the cause during hospitalization and the lack of effective treatment in case of inflammations of unknown etiology result in the extent and persistence of neurological symptoms [2]. Patients with a history of this disease, depending on their condition and commonly severe complications, often require constant neurorehabilitation to improve their functioning [3,4]. No official recommendations are available regarding physical therapy programs for encephalitis patients. There is a paucity of neuroscientific studies concerning the beneficial effects in patients with this disease. This paper presents the rehabilitation process of a patient with the diagnosis of encephalitis of unknown etiology, who was admitted to a rehabilitation clinic with three-limb palsy with preserved movement in the right ankle joint. The rehabilitation that was introduced allowed for the development and implementation of an individual communication model.

## 2. Materials and Methods

A 28-year-old female was admitted to the Department of Neurology in the morning due to paresthesia in the left corner of the mouth and tongue, accompanied by the deterioration of the left limb efficiency. The interview revealed that the patient had been healthy so far, socially very active, moderately physically active; in the past (two weeks before the hospitalization), she had had an upper respiratory tract infection, without fever. The onset of the patient’s symptoms had been noted before the start of the SARS-CoV-2 pandemic, so SARS-CoV-2 infection was not considered as a diagnosis. She developed a generalized tonic-clonic seizure and a fever of 38.8 °C during hospitalization. The patient’s condition was assessed as generally severe. She was conscious, somnolent, with lowered palate on the left side, loss of muscle strength in the left limbs, decreased muscle tone and weakened tendon reflexes on the left side. No respiratory distress was noted. Her vital signs were normal. Computed tomography of the head performed on the first day of hospitalization did not reveal any focal changes.

Lumbar puncture was performed on the second day. Nonspecific features were obtained in the cerebrospinal fluid (CSF), including cytoses 211 cells in μL, protein concentration of 62 mg/dL; glucose 58 mg/dL. Lyme disease and toxoplasmosis were excluded based on the CSF examination. Ciprofloxacin, acyclovir and dexamethasone were used in the treatment. Computed tomography was performed on the fourth day after admission. The examination revealed new foci of diffusion restriction in the cerebellar vermis and the left thalamus. Due to the presence of inflammatory lesions in the brain, immunoglobulin therapy was started. The patient’s clinical condition deteriorated, and an increase in neurological deficits and respiratory failure occurred as a consequence of pneumonia. The patient was unconscious with tetraplegia, abolition of tendon reflexes, decreased muscle tone, and a bilateral Babinski sign.

The MRI of the brain, performed one month after the admission, showed a heterogeneous focal lesion in the right thalamus (25 × 23 × 21 mm), with weak diffusion restriction, centrally with slight swelling and mass effect in the form of the compression of the right ventricular system, and a discrete shift of the midline to the left by about 1 mm, without any features of pathological contrast enhancement. During hospitalization, the patient underwent physical therapy, which included position changes in the bed and passive exercises of four limbs performed according to neurophysiological methods and orofacial therapy.

On the 32nd day of hospitalization, the patient was diagnosed with status post encephalomyelitis of unknown etiology, with the symptoms of tetraplegia, decreased muscle tone and weakened tendon reflexes. The patient was conscious and answered closed questions, communicating by blinking her eyelids. The patient was referred to a rehabilitation center for patients requiring respiratory support with a ventilator.

## 3. Results

### 3.1. Examination for the Needs of Physiotherapy in the Rehabilitation Center

On the day of admission to the center, the patient was conscious, non-ambulatory and tetraplegic, with respirator-assisted breathing using a tracheotomy tube. The ability to function independently was assessed using the Barthel and Functional Independence Measure (FIM) scales [5]. These standardized scales used for the examination of functioning were conducted to provide information about the assistance required by the patient. She scored 0 on the Barthel and 39 in the FIM scale (including 13 in the motor domain and 26 in the cognitive domain) [6]. Patient awareness was assessed using the Glasgow scale [7], in which she obtained a score of 10, including eye-opening (4/4), verbal contact (5/5) and motor reaction (1/6). The Glasgow Coma Scale is a typical assessment tool, used by neurologists to examine patients with severe central nervous system injuries. The functional status, which is the most important for the needs of the present study, was assessed according to the International Classification of Functioning, Disability and Health (ICF). ICF, as a standard frame-work in rehabilitation, allows the selection of therapy that is adequate to the patient’s functional needs. It also facilitates communication among members of an interdisciplinary team [8,9]. The physiotherapeutic examination outcomes are presented below.

The participation level: The patient was oriented towards time, place and herself. She adequately answered closed questions by nodding and negating with the foot. The patient was very conservative and distrustful when working with a physiotherapist. The lack of drive and mood swings were observed. The patient was unable to perform any daily activities independently.The activity level: Eyelid gaps were slightly closed and the patient was unable to perform facial movement. The patient was not able to change position in any way independently. Maintaining a sitting position was impossible due to the orthostatic reaction.The structure and function level: The patient did not report any kind of pain. There were no disturbances in superficial and deep sensation. The patient declared no sense of smell or taste. The abolition of muscle strength occurred in the upper limbs and the left lower limb. Lower subluxation in the shoulder joints of both upper limbs was present. As regards the right lower limb, deep paresis of the distal part was observed. The muscle tone and strength were lowered in all limbs, except the muscles responsible for movement in the right ankle joint (2 on the Lovett scale [10]). The range of motion within the joints of limbs was normal, except for the joints of the hands and flexion contractures within the fingers were observed bilaterally.

### 3.2. Therapeutic Intervention

The patient underwent comprehensive physiotherapy (see the Appendix A for the time course of the study). The rehabilitation program was established by an interdisciplinary team, including the cooperation of a doctor, physiotherapist, neuropsychologist, neurological speech therapist and a nurse. Physiotherapy was focused on the implementation of the following objectives at the level of participation: enabling communication with the environment using movement in the ankle joint and adaptation to sitting in a wheelchair to participate in physiotherapy outside the ward and to leave the rehabilitation clinic building. The patient’s rehabilitation was carried out daily for 1 h 45 min in two therapeutic sessions (60 min in the morning and 45 min in the afternoon) 5 times a week over 12 months.

Physiotherapy included work on the function of the limbs and improving the stabilization of the trunk. Changes in the folding positions within the bed were continuously used to prevent pressure ulcers. Physiotherapeutic prophylaxis was carried out through passive exercises of three limbs to prevent muscle contractures and maintain the range of motion in the joints. Additionally, active exercises of the right lower limb and orofacial therapy were implemented, including relaxing the muscles in the face and head, training the muscles controlling the movement of the eyeballs, exercises increasing the range of motion within the temporomandibular joints, relaxing the muscles of the tongue and exercising its mobility, stimulating the affected muscles in the face through vibration and face mimic exercises. Respiratory acceleration training was conducted to improve respiratory function.

The patient gradually adapted to the sitting position, initially by increasing the angle of the headrest, then by sitting on the edge of the bed with support for her back and head. The final stage included adaptation to sitting in a wheelchair with head support, as shown in Figure 1.

Lower limb muscle electrostimulation was also carried out. To improve communication with the environment skills, the patient was taught how to use an alphabetical table installed in the patient’s bed (adapted to her functional skills). The board made it possible to write on the computer and mark pictograms with a computer mouse operated with the right foot, as shown in Figure 2.

After 12 months of comprehensive physiotherapy, the patient’s score on the GCS and Barthel scales did not change. As regards the Functional Independence Measure, the patient scored 44, including 13 points in the motor domain and 31 points in the cognitive domain. It shows an improvement in the patient’s cognitive functioning. The patient was still insufficient in terms of the respiratory function and continued ventilator therapy. The patient’s condition according to ICF was as follows:The participation level: The patient communicated through the movements of her right foot (nodding and negating), opening and closing the eyelids (for closed-ended questions), and writing and marking pictograms on the computer. Improvement in mental condition and increased motivation to work with therapists were observed. Nevertheless, the patient was still unable to perform every-day activities independently. Work was underway on installing computer software with a voice simulator and a cyber-eye, which would allow the patient to improve communication with the environment.The activity level: Much greater ability to open the eyelid slits wide and the noticeable tension of the muscles around the mouth were noted. The patient could sit in a wheelchair with support for about 45–60 min without the orthostatic reaction. The remaining skills of the patient remained unchanged.The structure and function level: The patient still did not report any kind of pain. No disturbances in superficial and deep sensation were examined. Muscle strength of the lower right limb muscles increased (2+/3− on the Lovett scale) in the knee joint and the muscles responsible for movement in the ankle joint scored 4+ on the Lovett scale. Muscle tone was lowered in all limbs. The range of motion in the joints of the upper and lower limbs was within the physiological norm.

The comparison of the examination results is presented in Table 1.

## 4. Discussion

Encephalitis leaves permanent damage to the CNS and a decrease in functional efficiency and the quality of life is observed in most patients [11]. The fact that many patients with encephalitis have persistent deficits indicates the need for ongoing rehabilitation care and the role of the physiotherapist in maximizing functional skills, minimizing functional deficits or adapting compensation strategies, helping to integrate into the community, and facilitating participation in personally important activities. Moreover, the physiotherapist can help to provide a safe environment, gentle stimuli to encourage the process of spontaneous recovery, and to understand and adapt the patients and their families to the new situation through education and training. A well-trained and experienced physiotherapist, as a part of a holistic, multidisciplinary team, facilitates daily activities and helps to improve the quality of life.

Despite a significant number of papers concerning the diagnosis and treatment of encephalitis, little information can be found about the methods of rehabilitation in patients with this disease. As a result, physiotherapy in patients with encephalitis is carried out mainly based on the therapists’ own experience; thus, it is not based on scientific guidelines. A few available studies contain information on the positive impact of physical rehabilitation on the functioning of patients with encephalitis, carried out in an interdisciplinary team with other specialists [11,12,13,14]. Orofacial therapy has shown positive results in the wider opening of the eyelid slits. As a result of daily upright standing, the patient was adapted to prolonged sitting in a wheelchair [15]. Maintaining the range of motion and increasing the strength of muscles that supply the ankle joint was achieved through exercise and muscle electrostimulation [16]. Despite the training of respiratory acceleration and standing upright, no improvement was achieved in respiratory efficiency, which was necessary to make the patient independent of respiratory support. Applied technology and directing physiotherapy towards functional goals enabled the patient to better communicate with the environment and increased her motivation to participate in further rehabilitation. From a clinical evaluation perspective, little progress seems to have been made. However, in this case, it was of great importance for the patient that she was able to communicate better with her relatives despite her medical condition. Assessment and therapy based on the ICF concept may improve the patient’s functional outcomes. Without the cooperation of a physiotherapist in this case, it would probably be impossible for the patient to use her foot to operate the mouse. It would be impossible to improve or even maintain muscle strength in a limb, and to restore it and, above all, it would be difficult to learn a new function without physiotherapy. The neurophysiological methods used, in contrast to traditional kinesiotherapy (exercises that do not operate on the return of function in everyday activities, but only the restoration of movement in a selected range), in this case, might prove to be insufficient for such a difficult new task of handling a mouse with the foot.

This work supports the importance of an adapted rehabilitation program to the aim of the patient with severe disabilities and encourages larger studies to confirm the beneficial effects of physiotherapy treatment for patients with encephalitis. Recognizing the goal of the therapy and pursuing it, together with the involvement of the entire therapeutic team (including the physiotherapist), is crucial in the treatment process. Establishing and enabling contact with the environment may be a higher goal than the desire for locomotion.

## 5. Conclusions

Methods of communication with patients with profound motor deficits, including speech disabilities, remain a challenge for medicine and require individual selection and implementation in conjunction with rehabilitation, depending on the patient’s functional abilities.In the presented case, comprehensive physiotherapy significantly improved the patient’s ability to communicate and participate in social life.In patients with deep motor deficits, individually selected adaptations, taking into account modern technologies, should be sought to enable them to better communicate with the environment.The rehabilitation of patients with a significant degree of disability should be carried out in an interdisciplinary team, with various types of medical professionals, including a physiotherapist, psychologist and speech therapist.The paucity of the literature on physiotherapeutic interventions and their effectiveness indicates the need to create and publish more research in this area.

## Figures and Tables

**Figure 1 brainsci-12-00688-f001:**
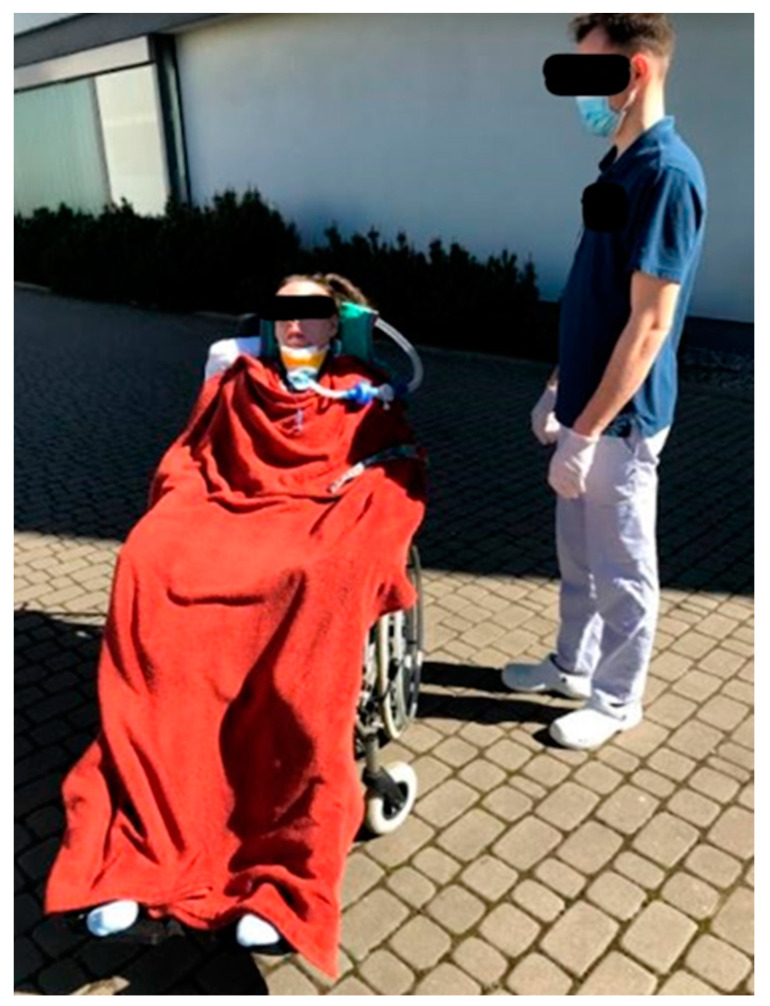
Patient adapted to sitting in a wheelchair.

**Figure 2 brainsci-12-00688-f002:**
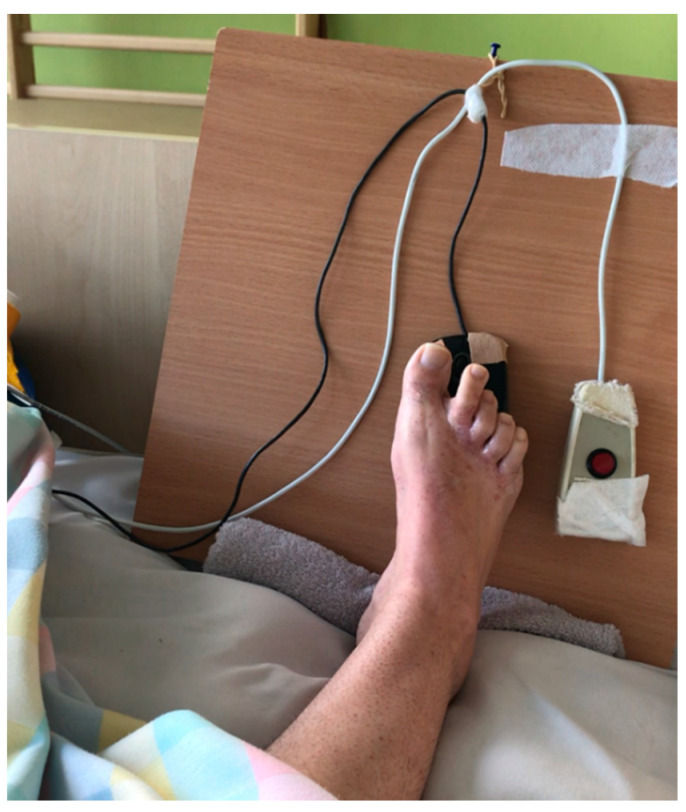
Patient’s ability to communicate by using the computer.

**Table 1 brainsci-12-00688-t001:** Comparison of the main factors included in the examination before and after the cycle of physiotherapy.

Level of ICF	Assessed Factor	Examination before Physiotherapy	Examination after Physiotherapy
**Participation**	communication	adequately answering closed questions by nodding and negating with the foot	communicating through the movements of the right foot (nodding and negating), opening and closing the eyelids (for closed-ended questions), writing and marking pictograms on the computer
	mental condition	conservative and distrustful when working with a physiotherapist	motivated to work with physiotherapists
	ability to self-manage	unable to perform any everyday activity independently	unable to perform any everyday activity independently
**Activity**	eyelid gaps	slightly closed	able to open the eyelid slits wider
	mouth opening	unable	noticeable tension of the muscles around the mouth
	changing and maintaining position	unable to perform any change of the position, unable to maintain the sitting position due to the orthostatic reaction	able to maintain the sitting position with the head supported for 45–60 min
	present pain	No	no
**Structure and function**	sensitivity	no disturbances in superficial and deep sensation	no disturbances in superficial and deep sensation
	muscle strength	paralysis, except muscles responsible for movement in the right ankle joint (2 on the Lovett scale)	the lower right limb muscle strength increased (2+/3− on the Lovett scale) in the knee joint, muscles responsible for movement in the ankle joint scored 4+ on the Lovett scale
	muscle tone	Decreased	decreased
	range of motion	range of motion in the joints of the upper and lower limbs within the physiological norm, except for the joints of the hands—bilateral flexion contractures within the fingers	the range of motion in the joints of the upper and lower limbs within the physiological norm

## Data Availability

Not applicable.

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
