# Peer review of "The Physiotherapy Process of a Plegic Patient Who Communicates with Foot Movement—A Case Report"

_brainsci, 2022, doi:10.3390/brainsci12060688_

Round 1

Reviewer 1 Report

The authors responded more of the comments made in the previous revision. However, authors should improve the quality of the discussion. Discuss about the effectiveness of the proposed treatment in this kind of patients. Compare the proposed therapeutic intervention with traditional approach in persons with sever disabilities.

Author Response

Response to Reviewer 1 Comments

Thank You very much for all your comments and suggestions. We have improved the manuscript in the abstract and the main text (additional/new information or discussion is highlighted in yellow, as follows in lines 10-12, 23-25, 38-40, 43-45, 188-197, 213-231, 246-247). In addition, the whole paper has been improved by the language professional.

Reviewer 2 Report

Thank you for conducting this interesting case study. The case studies generally consist in an in-depth investigation of a single individual, group, or event to explore causation in the hope of discovering the underlying principles. 

I think that this case study is needed to be considered against the following comment: 

1- please report the case study according to the CARE recommendations

2- please discuss in depth the potential role of PT in the management of the encephalitis

3- The pain is not Activity. Please recheck the ICF framework

4- what means that the PT may benefit the people with encephalitis. What is the role or the mechanism of PT 

5- I cannot find research or clinical recommendations in this study

Author Response

Response to Reviewer 2 Comments

Thank You very much for all your comments and suggestions. We have improved the whole manuscript (abstract and the main text), additional/new information or discussion is highlighted in yellow. In addition, the whole paper has been improved by the language professional.

Point 1: please report the case study according to the CARE recommendations

Response 1: The paper has been improved according to the CARE recommendations

Point 2: please discuss in depth the potential role of PT in the management of the encephalitis

Response 2: We have improved the introduction part and also the discussion (lines 188-197; 213-224).

Point 3: The pain is not Activity. Please recheck the ICF framework

Response 3: Thank you. Improved.

Point 4: what means that the PT may benefit the people with encephalitis. What is the role or the mechanism of PT 

Response 4: Thank you for that comment. As mentioned above we have improved the discussion (lines 188-197).

Point 5: I cannot find research or clinical recommendations in this study

Response 5: We added lines 225-231 and 246-247. We believe that there is a lack of research in this area (about the role of PT and physiotherapeutic possibilities) and this is important to highlight this issue.

Round 2

Reviewer 2 Report

The pain is still under activity in ICF 

Please report the study according to CARE recommendations

Author Response

Thank you very much for these comments. The "pain" mistake has been final improved in all places. Sorry for this omission.
I enclose the CARE checklist and additionally a timeline.

This manuscript is a resubmission of an earlier submission. The following is a list of the peer review reports and author responses from that submission.

Round 1

Reviewer 1 Report

In the presented case, comprehensive rehabilitation significantly improved the patient’s ability to communicate and participate in social life. 

1) does the patient sign any consent form?

2) please consider to uses a table or figure to show how the patient improved her functions. 

Reviewer 2 Report

Thank you for the opportunity of reviewing this interesting case presentation.

Please find down below some comments that may improve your paper.

Did you consider COVID encephalitis? I think that is important sice the patient had upper repsiratory tract infection, we are still in the pandemic and that this aetiology has been described (I Siow et al. Encephalitis as a neurological complication of COVID‐19: A systematic review and meta‐analysis of incidence, outcomes, and predictors. European Journal of Neurology. 2021.

Did you consider the use of FIM instead of Barthel (FIM includes evaluation of communication and social interaction).

Please revise the ICF concepts. On the othe hand, you can use the codes and qualifiers in order to demonstrate improvements.

Reviewer 3 Report

The present study corresponds to a case report or case study in which the authors describes the physiotherapy treatment of a 28-year-old female after acute encephalitis who was admitted to a rehabilitation center with “tri-limb palsy with preserved movement in the right ankle joint.

I consider it valuable that clinical teams try to document the rehabilitation process of their patients. This is undoubtedly a contribution to the rehabilitation sciences, however, in my opinion, this manuscript has important shortcomings that I describe below.

Major comments:

  1. I´m not agree with the classification or terminology used by the authors to describe the sensorimotor sequels of the patient after the encephalitis. According to my understanding, if the patients has only the right ankle joint movement preserved, we should classify the patient with a tetraplegia or with four limb palsy.
  2. The authors present Barthel scale and Glasgow coma scales as baseline assessment. Both scales seem unappropriated to classify the baseline cognitive and motor functioning of this patient (the patient was able to move only the right ankle). In fact, both clinical assessments did not change after 12 months of training.
  3. In addition to the previous point, the authors declare that “Informed consent was obtained from subject involved in the study. Written informed consent has been obtained from the patient to publish this paper”. How is that possible in a subject with Glasgow coma scale 10!!? See https://www.sralab.org/rehabilitation-measures/glasgow-coma-scale
  4. I´m not sure if the score in Glasgow scale is correct. The score was 10, however in the results section the authors mention that the patient was “oriented toward time, place and herself”. With this information, is not making sense that the score in Glasgow scale was 10 and not 15.
  5. The authors mention that he patient carried out speech therapy and psychotherapy. However, no results (pre vs. post) in objective assessment in those areas are reported. I recommend to add this information to complement the case report. For example, sign of depression and potential improvements in this area after the rehabilitation process.
  6. There are several scales to report participants´ sensorimotor (motor assessment scale, Function in sitting Test (FIST), Functional Independence Measure (FIM), trunk control test) and cognitive (Executive function performance test, Rappaport scale) functioning in sitting position or even in supine. Those scale would be more sensitive for this particular patient. In case the authors have those scales (or others) that are common in physical therapy teams please provide.

Minor comments

  1. More details regarding participant background is needed.
  2. Please clarify who sign the informed consent for this study.
  3. In the discussion-conclusion sections, the authors should summarize the principal characteristics of the therapeutic strategies used in this patient, and give their opinion about how this therapeutic approach contribute to the rehabilitation process of the case study presented in this manuscript, and how this approach could help other patients or other rehabilitation teams.